



# A Human-Portable Mass Flux Method for Methane Emissions Quantification: Controlled Release Testing Performance Evaluation

Coleman Vollrath, Thomas Barchyn, Abbey Munn, Clay Wearmouth, and Chris Hugenholtz

Department of Geography, University of Calgary, Calgary, Alberta, T2N 1N4, Canada

*Correspondence to*: Coleman Vollrath (coleman.vollrath1@ucalgary.ca)

**Abstract.** Quantifying methane ($CH_4$) emissions from anthropogenic sources is essential for compliance, inventory, and verification efforts. One established mass balance approach is the mobile flux plane method, in which emissions can be estimated from measurements of $CH_4$ at various heights. Most traditional applications require either a drone or aircraft, both of which can be expensive, or limiting to deploy in all situations. To broaden applicability and improve practicality, we adapted

the flux plane method using a long telescoping pole and backpack-mounted $CH_4$ sensor. We explored accuracy through 44 controlled release experiments (0.2–5.6 kg $CH_4$/h) from a 2.4 m stack. Measurements were taken at 6 heights (0.8–5.6 m) by walking transects 10–30 m downwind. The data yielded a mean relative error of -10.1%, with 68% of estimates within ±38.3% of true values. Results are comparable to drone-based methods. We also tested an optimized Gaussian plume model using partial vertical profiles to address issues where the pole could not measure the top of the plume. This approach was slightly

less accurate than the flux plane method but had less bias. Overall, results show that multi-level, telescoping pole-based $CH_4$ measurements combined with flux plane or Gaussian models can quantify emissions from elevated sources in a logistically practical manner with results comparable to other widely used mobile quantification methods.

## 1 Introduction

Methods to quantify anthropogenic methane ($CH_4$) emissions have become increasingly important recently as efforts to reduce

emissions have expanded. Abating $CH_4$ emissions is an effective method to mitigate global warming as $CH_4$ is a potent and short-lived greenhouse gas. The bulk of this method development has occurred within the oil and gas industry, driven by: (i) recognition that the oil and gas industry—a large emitter of $CH_4$—is emitting more than previously estimated (Howarth et al., 2011; Alvarez et al., 2018); (ii) cost-effective and high deployment readiness of mitigations (Ocko et al., 2021); (iii) regulations (e.g., EPA, 2023; European Commission, 2022; Government of Canada, 2023); and (iv) voluntary commitments by

governments and the oil and gas industry to reduce $CH_4$ emissions or credibly demonstrate the climate performance of natural gas. These factors and developments, alongside rapid advancements and innovation in $CH_4$ sensor technology, have nurtured the development of an expansive toolkit of quantification methods covering a variety of sources and rates in the oil and gas industry at different spatial and temporal scales (Vollrath et al., 2024).



Mass balance is a quantification method that uses directly measured $CH_4$ concentration and local wind measurements to evaluate the integrated $CH_4$ flux over a measurement plane or path (Karion et al., 2013; Conley et al., 2017; Corbett and Smith, 2022; Mohammadloo et al., 2025). The typical approach involves measuring $CH_4$ at various elevations (from ground level to above the plume), transecting the plume in a crosswind manner. With these data, the full cross-section of the plume can be approximated, and total flux can be estimated by multiplying the approximated concentration data by advection (wind speed) and integrating across the full cross-section. $CH_4$ can also be measured at a single elevation, typically in very far downwind applications such as basin-scale quantifications (Karion et al., 2013; Peischl et al., 2018), and careful assumptions made about the atmospheric boundary layer to assume the plume is homogenously mixed vertically.

Provided $CH_4$ measurements can be obtained across the full extent of the plume, mass balance methods have major advantages over atmospheric dispersion modeling. Atmospheric dispersion modeling often requires assumptions about atmospheric behaviour and in most cases is led by knowledge of the source location and characteristics. For large area sources, or complex situations with many sources with unknown rates and locations, traditional dispersion modeling can be extremely difficult. Difficulties with traditional dispersion models are exacerbated at close ranges and in situations where dispersion diverges significantly from theory and plumes are variable in form (Caulton et al., 2018).

The dominating practical challenge associated with mass balance methods is obtaining direct measurements of $CH_4$ across the full elevation of the plume. Plumes tend to vertically and laterally expand with downwind distance, becoming better mixed and extending higher into the atmosphere. The combination of this lateral and vertical expansion creates application specific constraints. Lateral dispersion and the desired spatial scale of measurement strongly affect where measurements must be taken. For example, measurement of individual oil and gas sites often cannot be meaningfully achieved 10s of kilometers downwind, as plumes from adjacent sites mix, introducing significant error into attribution and effectively limiting the spatial scale of the mass balance measurement. To measure individual equipment on oil and gas sites, data must usually be collected within 10s of meters downwind. Plumes at 10s of kms downwind can extend far above the surface, but close range plumes are much closer to the ground. Further complicating this is some release points of interest are elevated (e.g., flares, tank tops, etc.), meaning the plume disperses from an elevated point, both up and down within the atmosphere. These application constraints coupled with the typical geometry of plumes have led workers to use a broad diversity of platforms to address various spatial scales of measurement:

(i) Drones are highly suitable for measurements of equipment, typically at the 10s to 100s of m downwind range, enabling measurements of $CH_4$ at 100s of m elevation and close to the ground (e.g., Nathan et al., 2015; Tullos et al., 2021; Corbett and Smith, 2022; Dooley et al., 2024). This noted, drones differ. Fixed-wing drones often are limited in their ability to fly close to the ground compared to quadrotor drones (e.g., Nathan et al., 2015; Barchyn et al., 2017), limiting application to larger spatial scales where the bulk of the plume disperses upwards and the lack of measurements close to the ground are less limiting.



(ii)    Aircraft are best suited for measurements of larger facilities or groups of wells and facilities kms downwind (Conley et al., 2016; 2017; Lavoie et al., 2017; Li et al., 2024; Maazallahi et al., 2025). Similar to fixed-wing drones, aircraft cannot fly low. Other approaches involving the use of aircraft in extremely far downwind applications are noted (e.g., Karion et al., 2013; Peischl et al., 2018) but are largely inapplicable for finer spatial scales.

(iii)    Ground measurements (by foot, truck, or similar) are limited to site or finer spatial scales (measured 10s to 100s of m downwind). Ground access only allows measurement of the bottom portion of the plume. Efforts to get around this limitation have taken the form of vertical masts attached to vehicles (e.g., Thoma et al., 2015; Rella et al., 2015), but there are clear practical limits on the height of poles on vehicles associated with overhead obstacles on roads.

In the situation where only a portion of the vertical extent of the plume can be measured, a strategy is to augment the mass balance method with traditional plume models, creating a hybrid model that is more practical, but potentially carries larger error. Rella et al. (2015) used a Gaussian plume model to approximate unmeasured $CH_4$ above the reach of their vehicle-based pole. Similar issues occur with aircraft which face the opposite problem of being unable to measure the plume close to the ground. Traditional plume modeling (e.g., Caulton et al., 2018) is the endmember of this continuum where the vertical distribution of $CH_4$ is entirely modeled and measurements only exist near ground level.

The oil and gas industry has particular need for close-range methods that can accurately measure emissions from individual equipment on sites. Particularly problematic are emissions from high tank vents, flare stacks, or other elevated sources. Mass balance methods are well suited for these applications because close-range plumes tend to be poorly mixed and rarely conform to traditional plume model assumptions (Caulton et al., 2018), and there is a unique difficulty associated with performing measurements at close enough distances downwind (to accurately attribute emissions to individual pieces of equipment) but at sufficient elevations to capture the full plume (to reduce the need for models to predict unmeasured portions of the plume).

The dominant approach for these close range measurements is mounting a sensor on drones which are then flown 10s of m downwind. $CH_4$ emissions can be quantified using the flux plane method—an established mass balance approach—which involves projecting the measurements collected downwind of the source onto a two-dimensional plane oriented perpendicular to the wind direction (Nathan et al., 2015; Corbett and Smith, 2022). The plane is discretized into a grid across which the measurements are interpolated. This simulates a crosswind slice of the plume smoothed over time and space. Estimates of background $CH_4$ concentration and the local wind speed are then used to estimate the $CH_4$ flux passing through each discrete grid cell. Finally, the fluxes are integrated across the grid to estimate an emissions rate. Tests using quadrotor drones and the flux plane method exhibited a mean relative error of ±36.7% in non-zero controlled release experiments (Corbett and Smith, 2022). The ability of quadrotor drones to conduct vertical and close-to-source flights provides key advantages for emission quantification with the flux plane method. However, drones cannot be flown everywhere, and the cost and sophistication of





the miniaturized sensors remains a barrier to widespread application. Alternative or complementary methods that are simple, reproducible, easy to deploy, and accessible—while ensuring unbiased and accurate estimates—are warranted.

Here, to address the logistical issues associated with drones and advance methodologies associated with the flux plane method, we explore the use of a long pole, walked downwind of sources at various heights. This approach shows promise as it is logistically simple, comparatively safe, potentially low cost, and accessible to many researchers. We tested the full method in a series of controlled release experiments. We evaluated error to assess quantification accuracy and understand sources of error. To address situations where the pole cannot reach the top of the plume, we developed and tested a supplementary method that uses a customized forward Gaussian plume model. The results qualify the performance of the method for broader use in $CH_4$ emissions quantification.

## 2 Methodology

### 2.1 Test facility and release setup

We performed controlled releases of $CH_4$ on the 11, 12 of June and 6 August 2024 at the Carbon Management Canada (CMC) research site located approximately 20 km southwest of Brooks, Alberta (50.450510°, -112.120637°). We released natural gas from a compressed natural gas (CNG) trailer through a ~2.4 m tall release stack positioned in a large field. The terrain is flat with short- to medium-length grasses (0.2 m to 0.5 m) (see Fig 1). Gas was preliminarily regulated, then flowed to an ambient heat exchanger, which brought the temperature of the gas closer to the ambient atmosphere. An Alicat Scientific mass flow controller (MCR 2000SLPM-D) was then used to set and maintain release rates. The specification accuracy of the flow controller is: ± (0.8% of reading + 0.2% of full scale). During installation, release components were inspected using olfactory detection, personal multi-gas monitors, and an optical gas imaging (OGI) camera to identify any leaks. Wind speed and direction were measured with an RM Young 81000 3D sonic anemometer and data logger at 10 Hz a few meters away from the release stack at a height of 2.2 m. The locations of the release stack and anemometer were recorded with a survey-grade GPS.



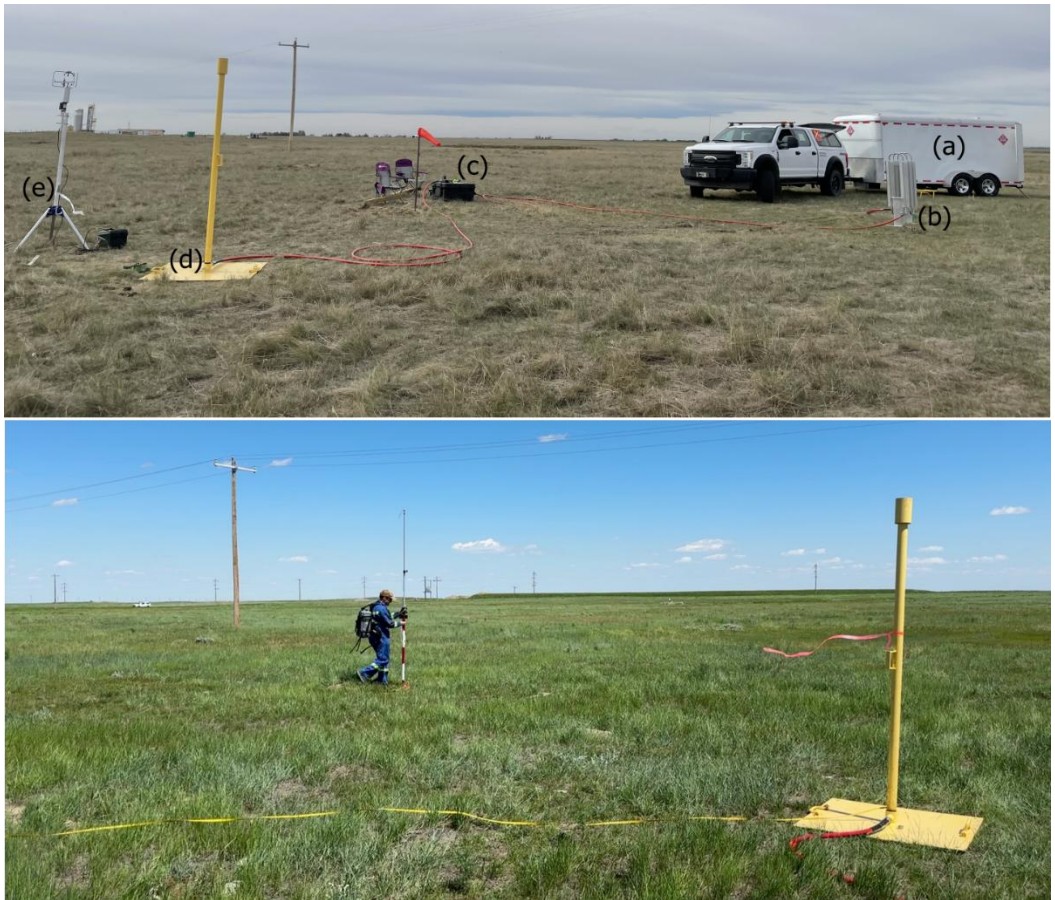

**Figure 1. Top panel—Controlled release setup. (a) CNG trailer with tanks. (b) ambient heat exchanger. (c) flow controller. (d) release stack. (e) anemometer. Bottom panel—the pole-based measurement system in operation during a controlled release test on 12 June 2024. The operator carried a CH₄ sensor in a backpack harness and a long length of hose was run from the sensor's inlet to the end of the pole where air was drawn in. Photo credits Zhenyu Xing and Chris Hugenholtz.**

Non-target sources of potential $CH_4$ emissions were screened using the University of Calgary's PoMELO vehicle system (Barchyn and Hugenholtz, 2020; 2022) to confirm that methane from unexpected upwind sources were not present during testing.

Gas samples were extracted from the CNG tanks to determine the average $CH_4$ content to correct the measured rates from bulk gas to $CH_4$. Samples were retrieved by creating a pneumatic trap in a bucket of water, which used gas flow to fill

an evacuated syringe and limit contamination from air. Samples were transferred to septa bottles for gas chromatography analysis. Additional details are provided in the Supplemental Material (SM1). Samples indicated released gas had an average $CH_4$ content of 90.5%.



## 2.2 Measurement kit and instrumentation

The pole-based measurement system developed for this test campaign consisted of a LI-COR 7810 (LI-7810) trace gas analyzer
in a backpack harness, an iPhone 14 Pro with Avenza Maps for GPS, a ~4.6 m SECO extendable aluminum survey pole, ~5.5
m of Teflon hosing, and a fanny pack (Fig 1 bottom panel). The hosing was connected to the LI-7810 inlet with a Swagelok
fitting and run to the top of the pole, such that the height of the hose inlet changed as the pole was repositioned or extended.
Although the analyzer was near ground level, the sample inlet was located at the top of the pole. The maximum practical height
of $CH_4$ measurement was 5.6 m, the minimum height could be within cms of the ground. The iPhone 14 Pro was mounted to
the pole using a bicycle cellphone mount such that the pole operator could view $CH_4$ mixing ratios measured by the LI-7810
in real-time and to record GPS location.

The LI-7810 measures $CH_4$, $CO_2$, and $H_2O$ vapor at 1 Hz; it has a measurable range of 0 to 100 ppm $CH_4$ with a
precision ($\sigma$) of 0.6 ppb at 2 ppm over 1 s (LI-COR, 2025). The LI-7810 pulls air into its sensor cavity through an internal
pump at a rate of 0.25 L/min. Air transit from the end of the hose to sampler creates a time delay, which was measured using
two methods. First, a small volume of bump gas (2.5% $CH_4$) was released near the end of the tubing with the LI-7810 running.
Second, we preformed a "$CO_2$ breath test," where a participant exhaled near the end of the hosing. A stopwatch was started
upon the participant releasing the gas or exhaling and stopped when the LI-7810 detected enhanced $CH_4$ or $CO_2$. Both methods
produced comparable results (16.4 s vs. 16.1 s) upon averaging a series of five tests. To minimize interpolation in subsequent
time-matching (and associated data smoothing), we used a round number 16 s time offset.

The iPhone 14 Pro is equipped with a precision dual-frequency GPS. We monitored GPS accuracy and estimated the
average horizontal spatial accuracy of the iPhone's GPS to be ~3 m. Note that there is no sky obstruction at the CMC test
facility to limit or vary position accuracy throughout experiments. Location data were logged every 1 s to match the frequency
of the LI-7810. Measurement height of the data was populated in post-processing (see details in Sect. 2.5.) and is expressed as
heights relative to the surface.

## 2.3 $CH_4$ release rates

We performed a total of 44 controlled release experiments over 11, 12 June and 6 August 2024. Release rates varied between
0.2 to 5.6 kg $CH_4$/h. This range was chosen such that the testing covered the anticipated $CH_4$ emissions rates of a variety of oil
and gas related sources, including natural gas distribution facilities (Lamb et al., 2015) and lower-emitting oil and gas
production sites (e.g., see Fig 8a in Vollrath et al., 2024). The release rates tested in this study were not representative of higher
emitting oil and gas production sites. We performed 2 replicates of the following rates 1.8 kg/h, 2.6 kg/h, and 3.3 kg/h.
Replicates were performed on different days to vary atmospheric conditions. Two experiments at different downwind
distances—either 10 m and 20 m or 15 m and 30 m—were performed for each release. See SM Table 1 for full details.





## 2.4 Data collection procedure

Measurements were made in a semi-circle around the release stack. This ensured that plumes were fully transected during each pass while minimizing the need to "chase" the plume around (i.e., walk further due to plume meander, as may be required with straight-line approaches) and walking distance. Additionally, standardizing the measurement distance enabled exploring the potential correlation between measurement distance and quantification error. We used a 50 m tape to flag distances of 10 m, 15 m, 20 m, and 30 m. The operator walked the flagged routes with the LI-7810 and telescoping pole to transect the plume. For each experiment, the release rate was set and surveys commenced after a ~3 minute delay to allow plume development. All surveys involved measurements both in the plume and in adjacent, non-plume atmosphere. Measurements outside of the plume are necessary to approximate the background $CH_4$ concentration.

During surveys, the operator oriented the telescoping pole such that the inlet of the tubing was facing the release stack. $CH_4$ mixing ratios were monitored in real-time (with olfactory detection of mercaptan, $CH_3SH$) to confirm the plume had been transected successfully. After passing the plume, the operator generally continued to the endpoint of the semi-circle to ensure the plume was fully captured horizontally and that background $CH_4$ was sampled. In some cases, the operator walked further than or stopped short of the ends of the pre-marked routes depending on where the plume intersected the semi-circle to improve sampling efficiency or collect additional data to use in estimating background $CH_4$. The operator then reversed the transect. This procedure was performed for a total of six measurement heights, yielding two transects (there and back) for each height (i.e., 12 plume passes). Figure 2a shows the semi-circle transect pattern for experiments at 15 m and 30 m downwind of the stack during a release of 2.4 kg $CH_4$/h, and Figs 2b and 2c show measured $CH_4$ mixing ratios as a function of increasing measurement height during these experiments.





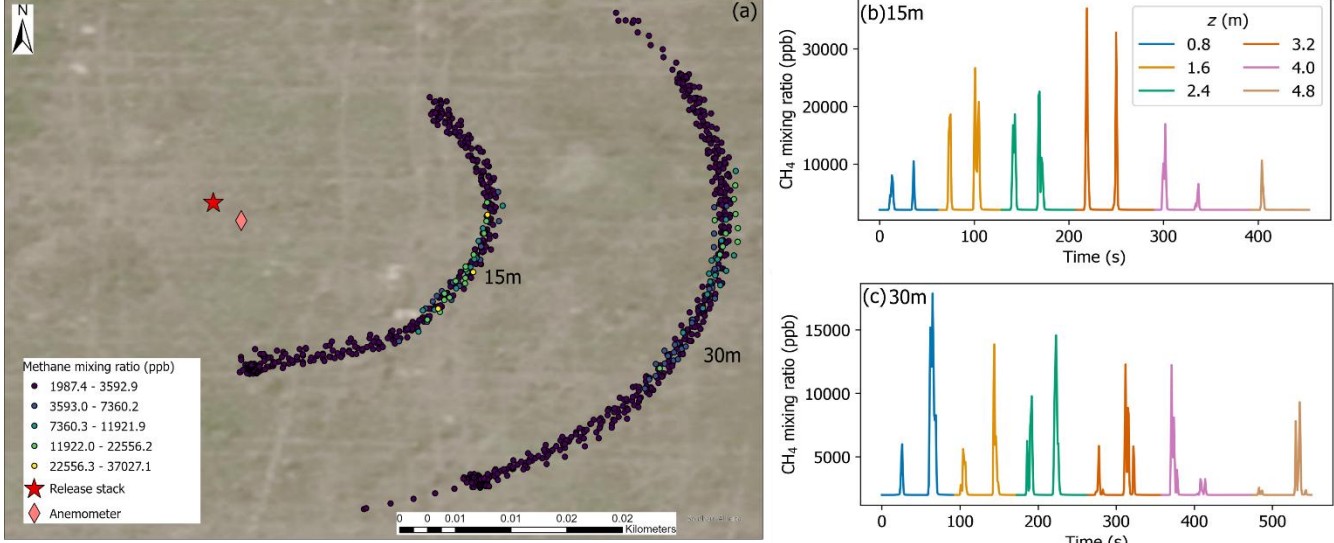

**Figure 2. (a) semi-circle transects walked with the backpack-mounted CH₄ sensor and telescoping pole at 15 m and 30 m downwind of the controlled release stack during a release of 2.4 kg CH₄/h on 12 June 2024. Circle colors correspond to CH₄ mixing ratios in the legend. (b) and (c) CH₄ mixing ratios by measurement height during the experiments at 15m and 30m downwind distances. The legend in (b) corresponds to measurement height by color in (b) and (c). Each group of two peaks corresponds to two passes of the plume at a single measurement level. Low or no visible enhancements towards the end of the experiments at higher measurement heights suggests the highest measurement heights captured the vertical extent of the plume.**

The measurement heights varied slightly between experiments. On 11 June 2024 the measurement range was ~0.8 m to 4.7 m with ~0.8 m spacing between measurement levels. On 12 June 2024 experiments were similar but had a slightly higher top measurement level of 4.8 m due to some improvements to methods to support the pole, facilitating a higher top height. On 6 August 2024, the top measurement height was 5.6 m with approximately 1 m spacing between heights. The release stack height of ~2.4 m suggested that the maximum height of the pole measurements was 2 to 2.3 times higher.

**2.5 Data preprocessing**

Data preprocessing began by correcting any measured time offsets between the LI-7810 and GPS data. Then, the static correction (16 s) for delay associated with air transit was applied. The GPS and LI-7810 data were fused, joining by time. Measurement height was attributed to the data based on the start and stop times of the measurements at each height. The angles and distances between the release stack and all measurement points were calculated. The spatial locations of the points ($x, y$) were plotted in cartesian coordinates where the release stack was located at (0, 0). Finally, we visually examined the plotted measurement locations, trimming the data in some cases where the vast majority of measurements were in non-plume



atmosphere. This enabled defining flux planes that were better fit to the scales of individual plumes, while still including non-plume data for background $CH_4$ estimation.

## 2.6 Flux plane method

### 2.6.1 Procedure to define the plane and project and interpolate the data

The flux plane method involves projecting the downwind $CH_4$ measurements onto a two-dimensional plane oriented perpendicular to the wind direction. $CH_4$ molecules are assumed to be advected through the plane by the wind. This simulates a crosswind slice of the plume and addresses potential issues with "off-axis" $CH_4$ measurements, which could distort the plume's shape and thus affect the estimated flux. There is, however, limited guidance in the research literature on determining the horizontal extent of the flux plane (Nathan et al., 2015; Rella et al., 2015; Corbett and Smith, 2022). Therefore, after

preprocessing the data, we defined the horizontal extent of our flux plane as the distance between the maximum and minimum $(x, y)$ measurement locations (orange points Fig 3), such that relative to the wind direction, the flux plane is initially situated in front of most measurement points downwind (blue line in Fig 3). We then adjusted the position of the flux plane using the average of the measurement locations (yellow diamond Fig 3) as the plane's (black line Fig 3) midpoint, as this minimizes the projection distance for the points onto the two-dimensional plane (Nathan et al., 2015). Higher data density at the starting point

of the semi-circle—due to the operator adjusting the height of the pole at this location after each set of two consecutive plume passes—may have influenced the point's location and is an area for method improvement. We rotated the plane to be perpendicular (green line Fig 3) to the reference wind direction/plume centerline, which we defined as the angle between the release stack and the peak $CH_4$ location in each experiment (Caulton et al., 2018).





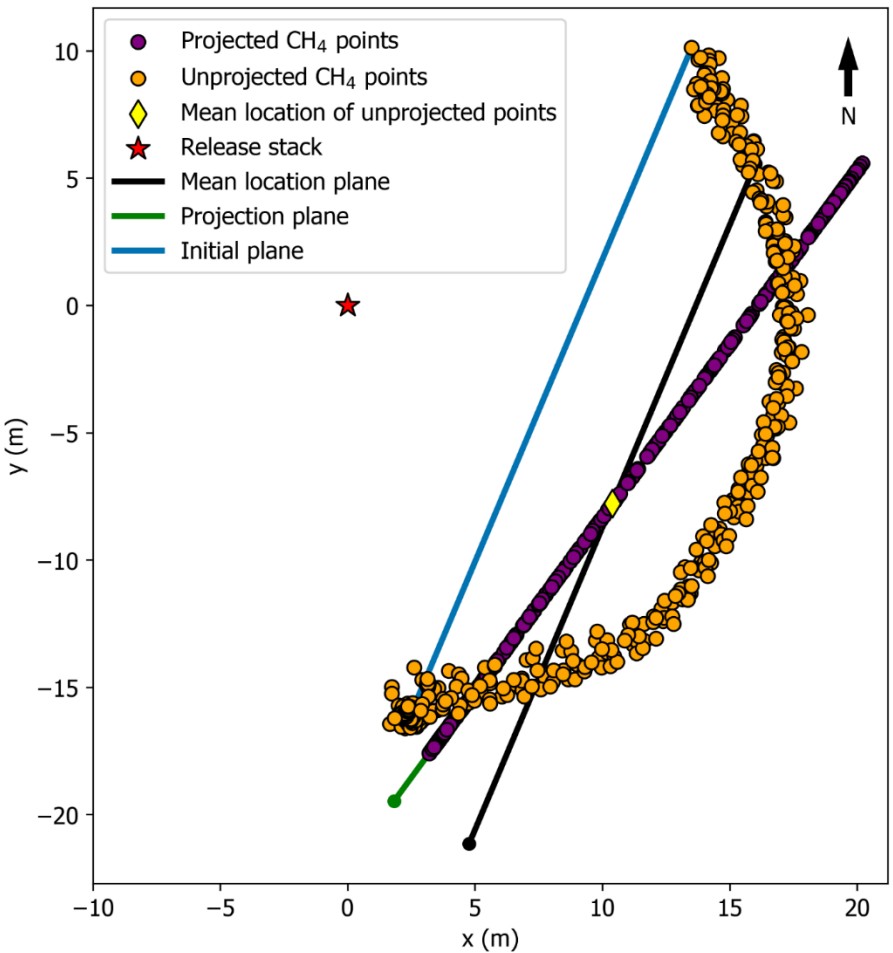

**Figure 3. Flux plane definition and projection of CH₄ measurement points within a cartesian coordinate system for a 2.4 kg CH₄/h release on 12 June 2024 ~15m downwind. See the associated description of the procedure to define the flux plane and project the data in the text.**

After defining and locating the flux plane, we projected the measurement points onto the plane (purple points Fig 3) using the dot product of the unprojected points (orange points Fig 3) and the plane. The line of projected data points was sometimes slightly shifted spatially on the plane that we defined depending on the spatial extent and angle of the plane relative to the unprojected data points (Fig 3). Therefore, we used all projected data points and their spatial extent in our quantifications instead of only using projected points that fell within the bounds of our defined plane. Initially defining a slightly larger flux plane in the horizontal direction or addressing data density issues in calculating the plane's midpoint (yellow diamond Fig 3) would have mitigated this issue.



We then calculated the crosswind distances for each projected measurement point. The vertical extent of the flux plane was set from 0 m (i.e., ground level) to the top measurement level plus half the distance between the top two measurement levels (Dooley et al., 2024). The total height of the plane was between 5m to 6.3 m depending on the experiment (Sect. 2.4.).

The vertical extent of the plane was set to be larger than the range over which $CH_4$ mixing ratio was measured to extrapolate the lowest measurements down to the surface and the highest measurements slightly above the highest measurement height. This is an extrapolation of measured data, but helps ensure that each measurement is more equally weighted in the total flux estimate.

We discretized the plane into a grid to estimate the $CH_4$ flux for each individual grid cell. The grid cell resolution was

set to 0.5 m$^2$. This resolution was generally about one-half the vertical spacing of the $CH_4$ measurements, enabling interpolation of $CH_4$ between the different measurement heights. The rate estimates were not overly sensitive to changes to the grid cell resolution (2.9%, see SM2 for a sensitivity analysis). Although, it should be noted that increasing the grid cell resolution beyond the scale at which the measurements were collected (e.g., ~1m in this study) may result in oversmoothed interpolated plumes, artificially inflating the estimated rates. We used an Inverse Distance Weighted (IDW) squared method to interpolate

measured data to the grid cell centers:

$$Z(P) = \frac{\Sigma_{i=1}^{n} \frac{Z_i}{d_i^2}}{\Sigma_{i=1}^{n} \frac{1}{d_i^2}} \tag{1}$$

where $Z(P)$ is the interpolated estimate of $CH_4$ mixing ratio at point $P$, $Z_i$ is the measured $CH_4$ mixing ratio at a nearby point

$i$, $d_i$ is the distance in meters between interpolated point $P$ and measured point $i$, and $n$ is the number of measured points used to interpolate a value at $Z(P)$.

SM3 explains our choice of IDW squared as the interpolation method. Figure 4a shows the interpolated $CH_4$ flux plane for the 2.4 kg $CH_4$/h release on 12 June 2024 at 15 m downwind. It should be emphasized that Fig 4a is a spatially and temporally smoothed representation of the plume measured in this experiment, simulated with interpolation.






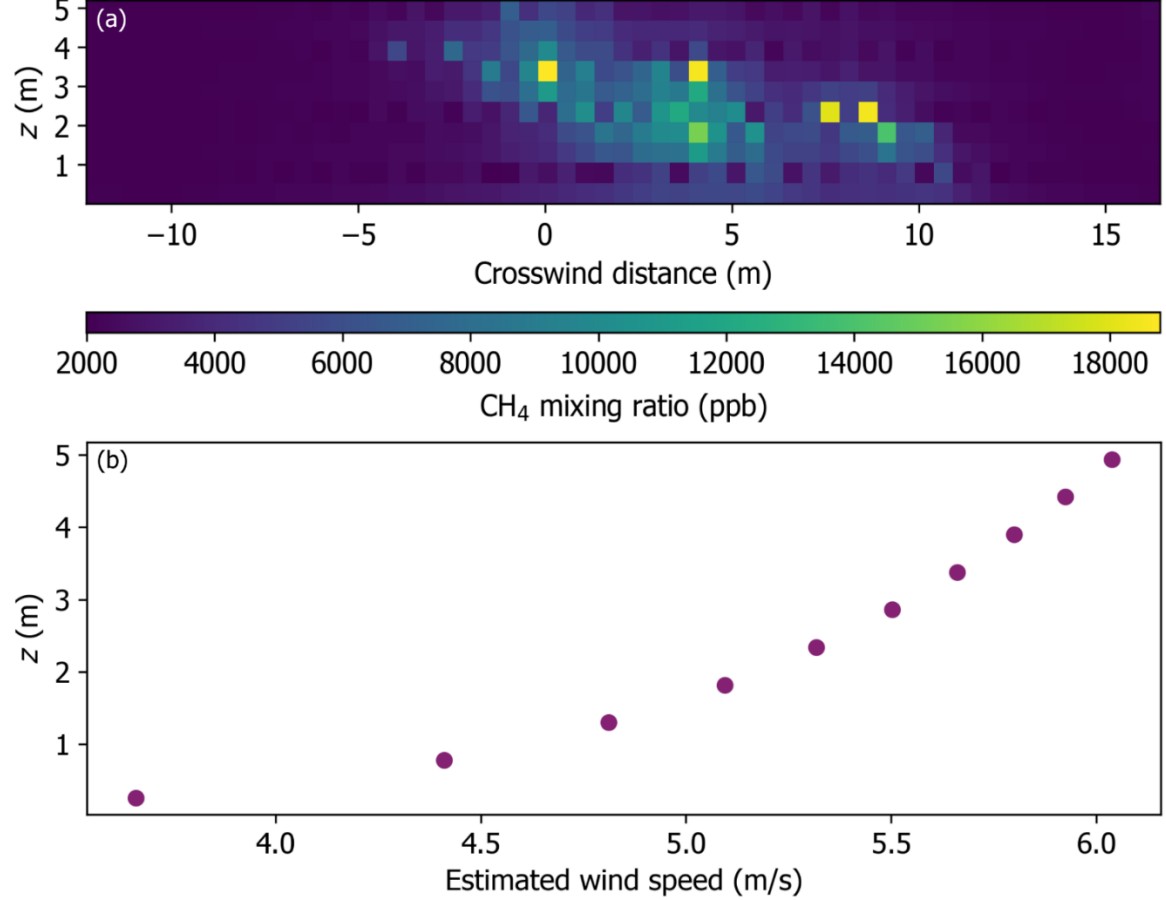

**Figure 4. (a) Interpolated flux plane of CH$_4$ mixing ratios for the 2.4 kg CH$_4$/h release at 15m downwind distance on 12 June 2024,**

**and (b) power law estimated vertical wind profile during the same experiment.**

### 2.6.2 Vertical wind speed profile estimation

We measured wind speed at a height of 2.2 m during the experiments. As such, it was necessary to extrapolate the wind speed measured at this height to match the heights of the interpolated CH$_4$ mixing ratios for emissions quantification. A power law

(Heier et al., 2014) was used to estimate wind speed for grid cell centers:

$$v_z = v_m \left(\frac{z}{z_m}\right)^a \tag{2}$$

where $v_z$ is the unknown wind speed in m/s at height $z$, $v_m$ is the mean wind speed in m/s measured during the experiments,

$z_m$ is the height at which wind speed was measured, $z$ is the extrapolation height, and $a$ is the wind shear coefficient.





For $v_m$ in Eq. (2), we used the mean wind speed measured during the release, aligned with recording time. Drift in the data logger clock relative to the iPhone was negligible (< 1s). The wind shear coefficient, $a$, was set to 0.17, which was empirically derived in similar controlled release testing (Rella et al., 2015). More discussion of $a$ is provided in SM4, although SM2 shows a lack of sensitivity of the quantified rates to different values of $a$. See Fig 4b for visualization of wind profile.

### 2.6.3 Flux plane quantification

After interpolating CH$_4$ mixing ratios onto the grid and estimating wind speed for the grid cell centers, we estimated CH$_4$ emissions rates for the release experiments using:

$$Q = \iint w_{gc} \left( \frac{(ch4_{gc} - ch4_{bg})}{10^6} \right) \rho_{ch4} res_{gc} t \qquad (3)$$

where $Q$ is a CH$_4$ emissions rate in kg/h, the integrals represent integration over the two dimensions of the interpolated CH$_4$ and grid, $w_{gc}$ is the estimated wind speed for each grid cell center, $ch4_{gc}$ is individual CH$_4$ grid cells, $ch4_{bg}$ is background CH$_4$ mixing ratio during the experiments, $\rho_{ch4}$ is the density of CH$_4$ estimated for the time of the experiments using the Ideal Gas Law, pressure and temperature data, and the molar mass of CH$_4$, $res_{gc}$ is the resolution of a grid cell in m$^2$, and $t$ is 3600s. We used the fifth percentile of CH$_4$ mixing ratio during the experiments for $ch4_{bg}$.

Hourly atmospheric tower measurements of pressure and temperature from Environment and Climate Change Canada (ECCC) for Brooks, AB were used to calculate $\rho_{ch4}$ (World Meteorological Association (WMO) ID: 71457).

### 2.6.4 Quality assurance

We developed a quality assurance protocol to identify surveys that are unlikely to produce reasonable results. Quality assurance protocols are essential in application to provide unambiguous guidance for subsequent field application of the method.

First, we screened all experiments to confirm that the measured peak CH$_4$ mixing ratio was below the top measurement level. All 44 (100%) out of the 44 controlled release experiments passed this criterion. This criteria provides indication that the majority of the plume was sampled and was not higher than sampled data.

To evaluate the geometry of the experiment, and better understand whether measured data captured the entirety of the plume, we estimated a Pasquill-Gifford (PG) stability class at the time of the experiments using the U.S. EPA's Key to Solar Radiation Delta-T Method (EPA, 2000). This method requires using 10 m wind speed, solar radiation, and time of day to look up a PG stability class from a table. We estimated the 10 m wind speed using the approach outlined in Sect. 2.6.2. and retrieved estimated solar radiation for the location and time of the experiments from NASA's POWER Project data access viewer (NASA, 2025). We used the estimated stability class and the downwind distance of the measurements to calculate a vertical dispersion parameter, $\sigma_z$, for each experiment using the Briggs-McElroy-Pooler Stability Class Lookup Table (Briggs, 1973). For each experiment, we examined the CH$_4$ time series and recorded the height at which we measured the highest CH$_4$




mixing ratio. We used this to approximate the vertical location of the plume's centerline. We then added this height to the calculated $\sigma_z$ for that experiment and subtracted this value from the topmost measurement level, to estimate if the measurement plane captured the majority of the $CH_4$ plume. Negative values suggest that the highest measurement height may not have captured the full plume. This flagged 4 (9.1%) of the 44 experiments.

Third, we visually inspected the experiments flagged by the second criterion to determine if the measurements captured the majority of the plume. We inspected the data for (i) a discrete plume in the data centered around the peak $CH_4$ mixing ratio, and (ii) a decreasing trend in $CH_4$ enhancements with height, such that enhancements measured at the top height were less than one-half of the peak $CH_4$ enhancement. This signals that the majority of the plume was captured within the plane. It may not be necessary to measure in fully clean air above the plume to yield accurate rate estimates. One (2.3%) of the 44 experiments did not meet this criteria as $CH_4$ enhancements were less at the second-highest measurement level, then increased above one-half of the peak enhancement at the highest measurement level (see SM Fig 1 in SM5). This plume likely changed vertical location between measurement of the top two measurement heights. In this experiment, the mean wind speed was the lowest of all experiments across the three testing days (2.8 m/s), resulting in an estimated PG stability class of B. We did include this experiment in the dataset, but is separately discussed to help learn how these situations affect results (see SM5).

**2.7 Optimized forward Gaussian method**

The flux plane method requires that the majority of the $CH_4$ plume be captured within the measurement plane to accurately estimate emissions rates. In cases where the pole is insufficiently high to capture the majority of the plume, other methods are required. To address these situations, we explored the use of forward Gaussian plume model optimization on the data. This follows efforts from other workers— Nathan et al. (2015) applied a highly nuanced forward Gaussian optimization to confirm $CH_4$ emissions rates from a natural gas compressor station quantified with a fixed wing UAV and the flux plane method.

To test the forward Gaussian optimization method, we removed data from the top two measurement levels in 43 of the 44 experiments to simulate a situation where the pole can only reach the plume's centerline. This is equivalent to a 30% to 42% reduction in measurement height depending on the day of the experiment and the vertical capabilities of the kit on that day (Sect. 2.4.) or using a pole of about ~3.2 m to characterize the $CH_4$ emissions released from the ~2.4 m stack. To estimate emissions with this method, the plume's centerline should be measured at the top measurement height or below, enabling optimization of the plume's vertical dispersion coefficient, $\sigma_z$, based on measurement of the plume's bottom half. As such, for one experiment, we only excluded data from the top measurement level because the measured peak $CH_4$ mixing ratio occurred at an altitude of 4 m.

Figure 5a shows the 2.4 kg $CH_4$/h release at 15 m downwind where the peak $CH_4$ mixing ratio was measured at an altitude of 3.2 m; removing the top two measurement levels in this experiment simulated a situation where only the bottom part of the plume was measured.




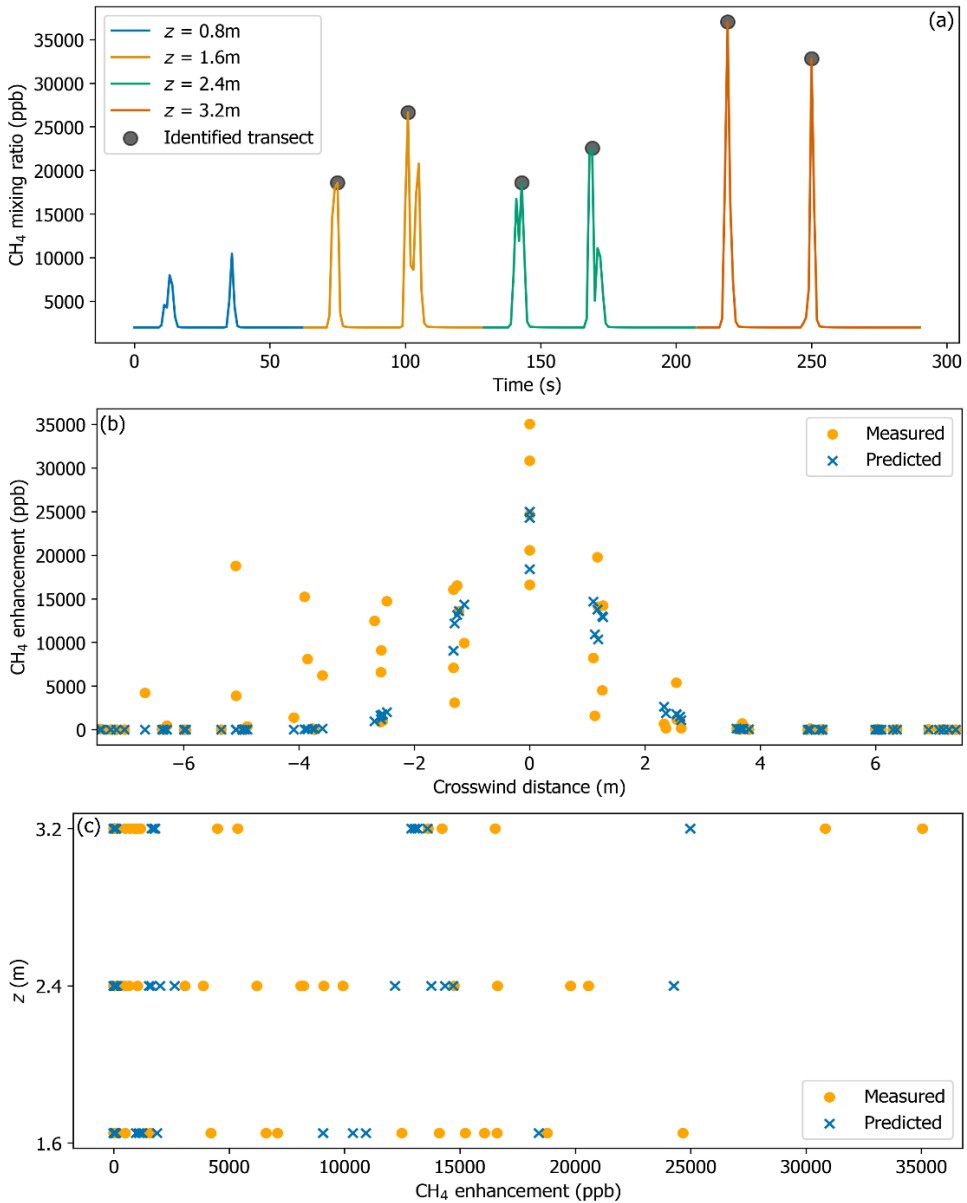

**Figure 5. Experiment on 12 June 2024 at a release rate of 2.4 kg CH$_4$/h and 15 m downwind measurement distance. (a) transects by measurement height. The top two heights (4.0m and 4.8m) were removed to simulate a situation where the pole could only measure CH$_4$ at or slightly above the plume's centerline. Identified transects were selected for further data processing (see associated description in the text) and used in quantifying emissions. (b) and (c) horizontal and vertical fits of CH$_4$ mixing ratios predicted by the forward optimized Gaussian model to measured mixing ratios. The optimizer estimated a CH$_4$ emissions rate of 3.1 kg/h, which is equivalent to a 28.6% error relative to the release rate.**





During algorithm development and testing, we noted a tendency of the forward Gaussian optimization method to substantially underestimate CH₄ emissions rates for some experiments when data from all measurement transects were used. We hypothesize

this was due to the close-range nature of the measurements and their use within the optimizer. For the majority of release experiments—particularly when measurements were performed at very close downwind distances (e.g., 10m to 15m)—only a few walking transects had high CH₄ enhancements consistent with the vertical location of the plume's core, while others often had much lower enhancements. Given that the optimizer solved for the global minima between measured and predicted CH₄ mixing ratios based on Root Mean Square Error (RMSE), transects with low CH₄ enhancements dominated the optimizations

when these were high in count, and predicted Gaussian plume peak concentrations were often much lower than measured. Thus, it was necessary to prioritize data from transects with high CH₄ enhancements to improve forward Gaussian model fits and rate estimates.

From test measurement heights, we identified transects with peak CH₄ mixing ratios that were greater than or equal to the sum of the mean CH₄ mixing ratio across all transects and two standard deviations (identified transects in Fig 5a) for

further data processing. We found this criterion was optimal for estimating rates, as it targeted transects consistent with the plume's core vertically. We then sliced data from these transects centered around the transect peak (grey points in Fig 5a) and spatially aligned the plume centerlines of each transect following the approach described in SM6. This corrected for changes in the wind direction across the transects that shifted the plume's location (Caulton et al., 2018). After alignment, we calculated the crosswind distance of the data points. The angle between the release stack and the location of the highest CH₄ concentration

measurement across all identified transects served as the reference wind direction/plume's overall centerline (Sect. 2.6.1.) for the calculation.

We used a forward Gaussian plume model with ground reflection (Eq. 4) with an optimizer to minimize the RMSE between predicted and measured CH₄ mixing ratios. Eq. (4) describes the forward Gaussian plume model:

$$C(x, y, z) = \frac{Q}{2\pi\mu\sigma_y\sigma_z} exp\left(-\frac{y^2}{2\sigma_y^2}\right)\left[exp\left(-\frac{(z-H)^2}{2\sigma_z^2}\right) + exp\left(-\frac{(z+H)^2}{2\sigma_z^2}\right)\right] \qquad (4)$$

where $C(x, y, z)$ is the CH₄ enhancement at location $x, y, z$ predicted by the model, $Q$ is the CH₄ emissions rate, $\mu$ is wind speed, $\sigma_y$ and $\sigma_z$ are the lateral and vertical plume dispersion coefficients, respectively, $H$ is the effective stack height, and $y$ and $z$ are the crosswind and vertical distances from the centerline of the plume.

To optimize we used the Limited-memory Broyden-Fletcher-Goldfarb-Shanno method, with Box constraints (L-BFGS-B) to optimize the following parameters: $Q$, $\sigma_y$, $\sigma_z$, and $H$. The method allowed bounding of $Q$ to prevent unrealistic rate estimates, which we set to 3.6 x 10⁻⁴ kg CH₄/h to 36 kg CH₄/h. Initial guesses were provided to the optimizer for $Q$, $\sigma_y$, $\sigma_z$, and $H$. We provided an initial value for $Q$ at 3.6 kg/h. Other initial parameters, $\sigma_y$, $\sigma_z$, were estimated based on the Brigg's stability class at the time of the measurement (Sect. 2.6.4.). The initial value for $H$ was the mean height of the identified

transects in Fig 5a weighted by CH₄ mixing ratio. We bounded $H$ in the optimizations to the initial value ± 0.5m, to minimize



potential fitting errors associated with plume meander vertically between the transects. Wind speed was estimated with the power law (see Sect. 2.6.2. for details) and we used this array of values for $\mu$.

Predicted $CH_4$ mass densities from the Gaussian model were converted to mixing ratios. Figures 5b and 5c show the horizontal and vertical fits of the $CH_4$ mixing ratios predicted by the forward optimized Gaussian model to the measured

mixing ratios for an experiment on 12 June 2024: release rate 2.4 kg $CH_4$/h and downwind measurement distance 15 m.

We evaluated the optimized values for $Q$, $\sigma_y$, $\sigma_z$, and $H$ and the horizontal and vertical fits of the predicted $CH_4$ mixing ratios to the measured data to identify potential issues requiring intervention and explore the model's performance. The optimizer converged on a solution to minimize the scalar (objective) function 100% of the time but predicted unrealistic estimates for $Q$ and $\sigma_z$ in 3 (6.8%) of 44 experiments. This was evident by the prediction of very large values for $\sigma_z$ that were

one to two orders of magnitude larger than expected based on the estimated stability class at the time of the experiment. In these cases, we set $\sigma_z$ to the optimized value for $\sigma_y$ and re-ran the optimization. Poor estimates of $\sigma_z$ by the optimizer were associated with experiments where plumes lacked a Gaussian shape in the vertical dimension, such those lacking a clear plume peak (e.g., high and similar $CH_4$ mixing ratios across the majority of transects then dropping sharply at higher altitudes), or those with notable meander of the plume's centerline. In one experiment the model predicted uniform concentrations across

the test measurement heights, possibly due to the irregular plume structure measured in this experiment (see SM Fig 1). In this case we fixed $H$ to the height at which the peak $CH_4$ mixing ratio was measured, improving the vertical fit.

## 3 Results

### 3.1 Testing envelope

Wind speeds across the 44 controlled release experiments on the 11, 12 of June and 6 of August 2024 averaged 4.8 m/s.

Minimum wind speed was 2.8 m/s and maximum wind speed was 7.9 m/s (Fig 6a). Mean wind speeds were < 3 m/s in two (4.5%) out of the 44 individual experiments; wind speeds were generally between 4 m/s and 6 m/s in the majority of experiments. The experiments were performed over a wide range of temperatures (16.8 ºC to 29 ºC) with a mean of 22.9 ºC (Fig 6b). The lack of continuity of the data in Fig 6b reflects the different temperatures on each testing day. The 6 August 2024 had the lowest temperatures while the 11 June 2024 had the highest temperatures. The total number of experiments performed

under the Briggs stability classes of B, C, and D were 2 (4.5%), 12 (27.3%), and 30 (68.2%), respectively. The mean survey time to complete the pole-based transects in the experiments was 9:35, varying from 6:20 for closer downwind distances to 16:15 for farther downwind distances (Fig 6c).





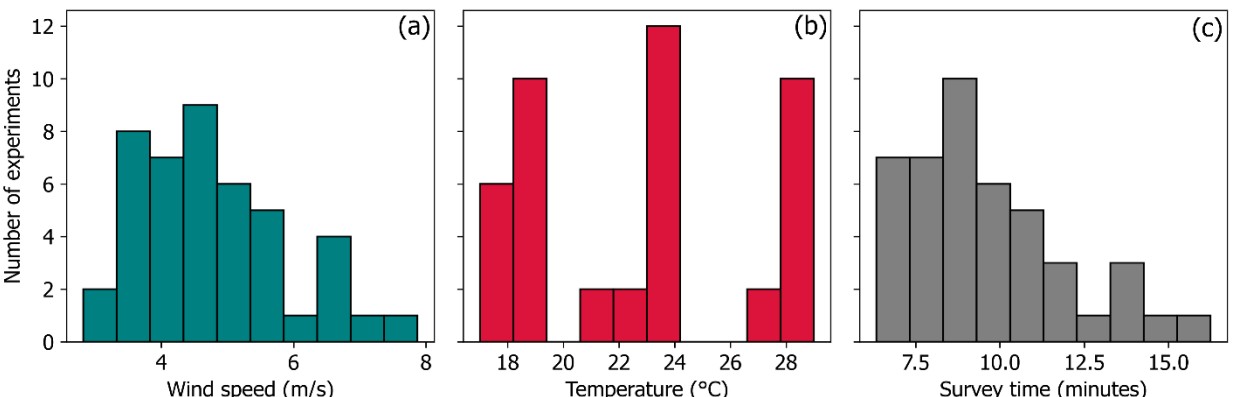

**Figure 6. Envelope of testing conditions for controlled release experiments on 11, 12 of June and 6 of August 2024. (a) wind speed, (b) temperature, and (c) survey time. Temperature data was acquired from the ECCC weather station for Brooks, AB (WMO ID: 71457).**

## 3.2 Flux plane quantification accuracy

We used Ordinary Least Squares regression in Python with a y-intercept of zero to compare estimated $CH_4$ emissions rates with the release rates. Figure 7 shows a fitted slope of 0.89 and an $r^2$ value of 0.96.

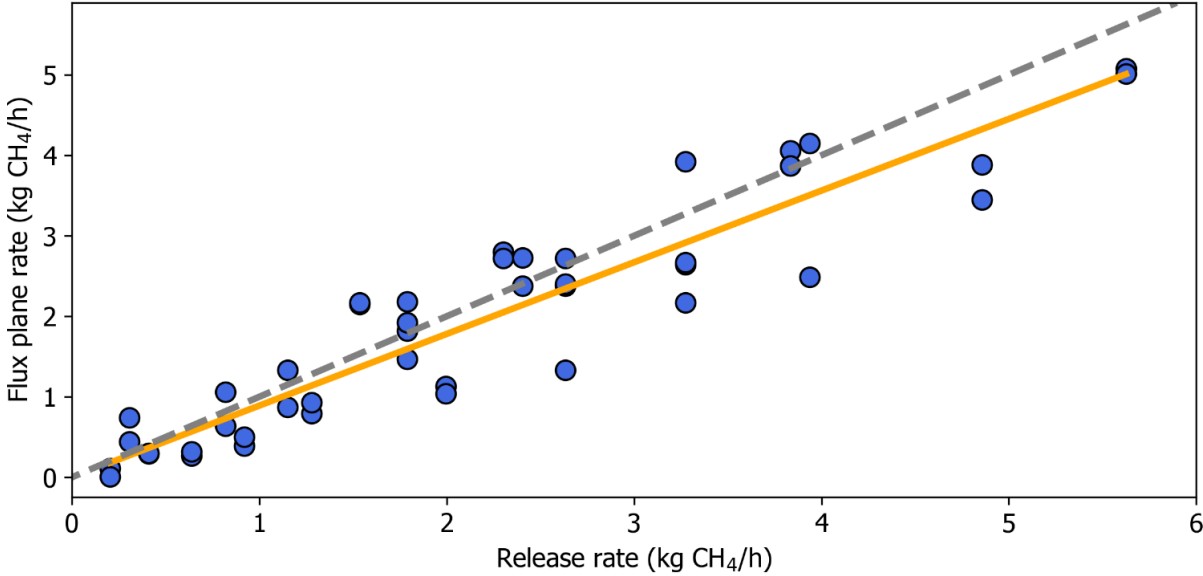

**Figure 7. Ordinary Least Squares (OLS) regression of flux plane estimated $CH_4$ emissions rates on the release rates. The orange line shows the model fit and the grey dashed line is 1:1.**



The mean percent relative error ((estimated rate – release rate) / release rate) * 100) of the CH$_4$ emissions rates estimated with the flux plane method was -10.1% (Fig 8a). Relative errors were mainly distributed around 0%, but a higher frequency of negative errors resulted in a left-skewed error distribution (Fig 8a), indicating a slight negative bias to the method. Approximately 68% of the quantified rates were within ±38.3% of the release rates. Minimum and maximum errors were -

97% and 141%, both of which were associated with experiments at the lowest (0.2 kg CH$_4$/h) and second lowest (0.3 kg CH$_4$/h) release rates, respectively.

Figures 8b to 8d generally show a lack of correlation between relative error and wind speed, release rate, and downwind distance. Figs 8b and 8c suggest (i) the flux plane method underestimated CH$_4$ emissions rates at higher wind speeds > 6 m/s, and (ii) larger errors were associated with lower release rates—errors decreased as the release rate increased. Figure

8d shows similar and fairly narrow error ranges within ±50% of the release rates across all downwind distances, although negative errors appeared more frequent at 10m downwind distance. More experiments are necessary to confirm these trends.

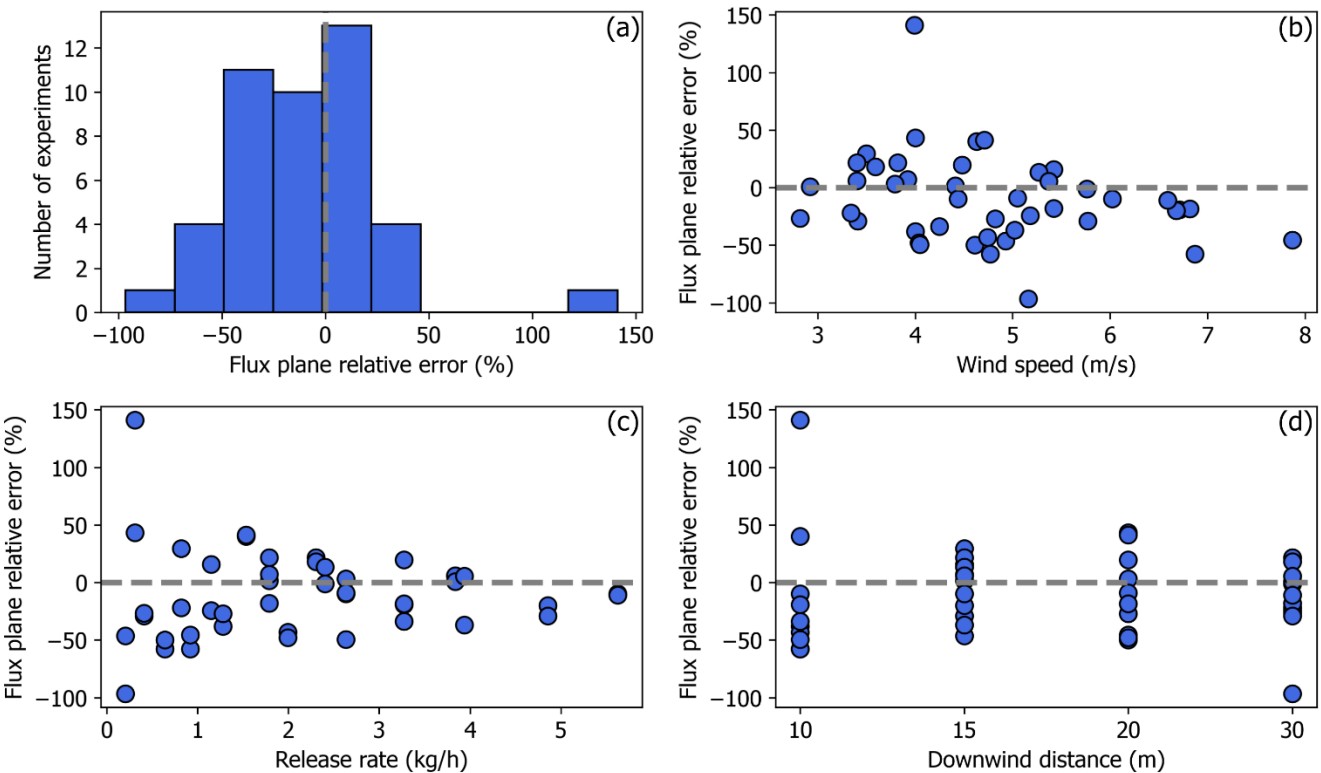

**Figure 8. (a) Percent relative error distribution of the CH$_4$ release rates quantified with the flux plane method. (b), (c), and (d):**
**percent relative error as a function of wind speed, release rate, and downwind measurement distance, respectively.**



### 3.3 Forward Gaussian optimization quantification accuracy

To test the forward Gaussian optimization method, we removed the upper measurement heights (30% to 42% of the data depending on the experiment). Figure 9a shows the estimated $CH_4$ emissions rates with the forward Gaussian method (linear model slope = 0.97, $r^2$ = 0.92).


One data point deviated substantially from the 1:1 line in Fig 9a. This corresponded with a 4.9 kg $CH_4$/h release at a measurement distance of 30 m on 12 June 2024. We measured nearly identical peak $CH_4$ mixing ratios at 0.8 m and 4 m altitude during this experiment (see SM Fig 3). Based on the vertical distribution of $CH_4$ mixing ratios, the majority of the plume appeared to be located closer to the ground. When removing the upper measurement heights to test the forward Gaussian

optimization method, the peak at 4m was removed. Given the deviation of this point from the 1:1 line, it is possible the irregular vertical shape of the plume and the removal of the peak at 4m was related to the underestimated $CH_4$ rate. The 0.3 kg $CH_4$/h release at 10 m downwind produced the highest relative error of the experiments with both the flux plane and forward Gaussian methods. Approximately 68% of estimates were within ±51.1% of the release rates. Errors ranged from -98.5% to 223%, which is similar to the flux plane method. Figure 9b shows a more even distribution of errors around 0% compared to the flux plane

method and suggests little bias to the method (2.4% mean relative error). Figures 9c to 9e show that the lack of correlation between quantification error and wind speed, release rate, and downwind distance are generally very similar to the flux plane method. Figure 9e suggests that the Gaussian method may not underestimate $CH_4$ emissions rates at 10m downwind distance as frequently as the flux plane method.




**Figure 9. (a) Ordinary Least Squares (OLS) regression of optimized forward Gaussian plume (FGP) model estimated CH₄ emissions rates on the release rates. The blue line shows the model fit and the grey dashed line is 1:1. (b) percent relative error distribution of the CH₄ release rates quantified with the forward Gaussian optimization method. (c), (d), and (e): percent relative error as a function of wind speed, release rate, and downwind measurement distance, respectively.**



Figure 10 shows that the CH$_4$ emissions rates quantified with the flux plane and the forward Gaussian methods are very similar despite removing the top two measurement levels from the latter for testing. Points do not deviate substantially from the 1:1 line. Forward Gaussian rates are generally larger, likely owing to the method's slight positive bias compared to the flux plane method's negative bias.

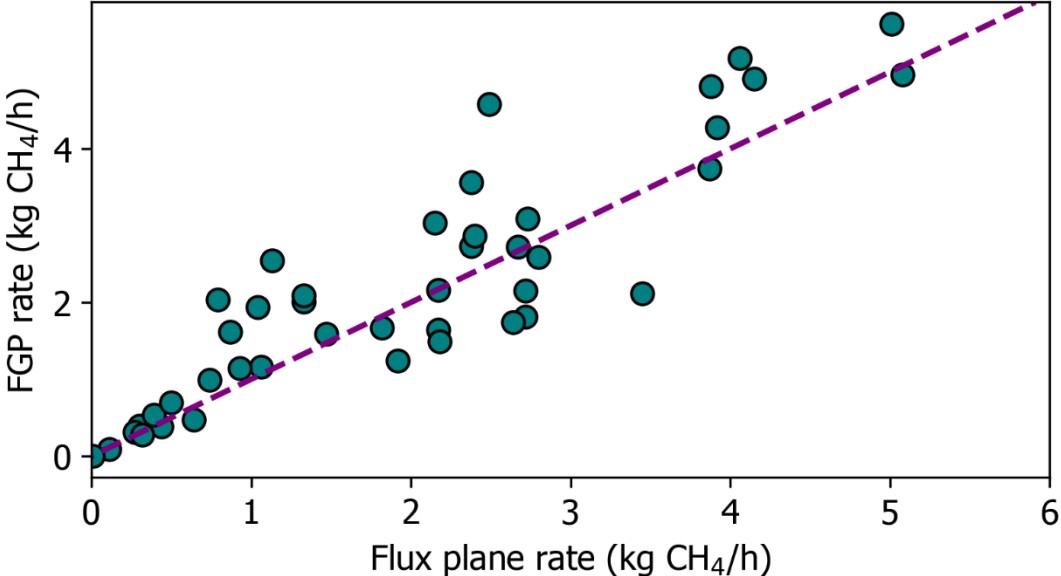


**Figure 10. Comparison of CH$_4$ emissions rates quantified with the flux plane and forward Gaussian plume (FGP) optimization methods. All CH$_4$ measurement data was used with the flux plane method. The forward Gaussian method was tested by removing the top two measurement levels (30% to 42% of the vertical plume profiles) depending on the test day and vertical capabilities of the measurement kit on that day.**


## 3.4 Replicate experiments

Some replicate releases were performed across different days for the rates 1.8 kg/h, 2.6 kg/h, and 3.3 kg/h. The initial releases were performed on 11, 12 of June 2024, the replicates were performed on 6 August 2024 at the same downwind distances. If all individual experiments at these rates are considered, percent relative errors ranged from -49.5% to 21.8% for the flux plane

method (-46.8% to 30.8% for the forward Gaussian method). We grouped the initial and replicate experiments by release rate and downwind distance and averaged the mean relative error of the paired experiments. This constrained the range of relative errors from -29.6% to 4.5% for the flux plane method (-40.4% to 7.0% for the forward Gaussian method). Despite the limited number of replicate experiments performed, the results suggest that performing replicates can improve the accuracy of CH$_4$ emissions rate estimates.



## 4 Discussion

### 4.1 Testing representativeness

The results are linked to the characteristics of the controlled release setup, environmental conditions, the measurement kit, and specific downwind distances for data collection. We used a telescoping pole with vertical capabilities approximately 2 to 2.3 times the height of the release stack (~2.4 m) to measure $CH_4$ mixing ratios at close downwind distances (10 m to 30 m). $CH_4$ was released from the stack at approximately ambient temperature with low exit velocity, the latter of which was related to the stack diameter (3" to 4") and release rates. No obstructions that could have modified the turbulent mixing of $CH_4$ were present between the stack and the end of the telescoping pole. Nearly all experiments were performed in wind speeds ≥ 3m/s. Overall, plumes had minimal buoyancy upon exiting the stack, and results suggest that this situation was successful at capturing the full plume in the majority of cases.

To achieve reliable results with the human-portable pole method, a worker must have *a priori* knowledge of the source location(s), and ideally have some indications of the plume behaviour during the experiment. One of the primary limitations of the method is that there are physical constraints to the height of a pole that can be operated by a person. The pole used in this study could be improved. We estimate that the maximum vertical capability of a human-portable pole-based measurement kit, when hardware is sufficiently optimized, could be near 8 m. A pole with vertical capabilities of 8 m should fully capture most plumes (even with some plume buoyancy) from sources approximately one-half of this height ≤ 30 m downwind. To fully capture plumes from taller (e.g., 5 m to 6 m) or slightly buoyant plumes, measurements in higher wind speeds could be a solution to minimize plume rise. The rapid deployment and operational flexibility of the method means that it is logistically much simpler to be selective with measurement conditions, likely enhancing measurement quality. From our experiments, there is an upper wind speed limit. It is more challenging to stabilize the pole when wind speeds ≥ 7 m/s. This limit could be mitigated by using lower, more stable poles, with consideration of how this affects experiments and workers.

Applications for the human-portable pole method include quantifying $CH_4$ emissions from oil and gas wells and facilities with shorter (≤ 6 m) equipment under moderate to high wind speeds, individual short- to medium-height equipment under similar conditions, natural gas distribution and metering and regulating facilities, abandoned wells, some landfill gas capture infrastructure, shorter wastewater pumping and lift stations, compost piles, and likely many other sources at closer downwind distances. Site access may be required to execute measurements of some sources at closer distances, but other sources may be publicly accessible for measurement, monitoring, or independent confirmation (e.g., Hugenholtz et al., 2021). Shorter telescoping poles can be used to target shorter sources under appropriate conditions. The pole-based iteration of the flux plane method is not an appropriate method to quantify $CH_4$ emissions from highly buoyant or lofted plumes due to stack exit velocity or temperature, such as compressor exhaust, high pressure line heater stacks without capped ends, or flare stacks. Drones provide a presently viable, but logistically more complex solution for these sources (e.g., Nathan et al., 2015).



## 4.2 Quantification accuracy

The human-portable flux plane method developed in this study using a telescoping pole to measure $CH_4$ emissions at close downwind distances estimated the release rates with a mean relative error of -10.1%. Approximately 68% of the estimated $CH_4$ emissions rates were within ±38.3% of the release rates.

One experiment deviated from the others (141% relative error), and examining the specifics of the release highlights important aspects of the method. For this experiment, $CH_4$ was released at the second-lowest rate of 0.3 kg/h, and measurements were taken at 10 m downwind. SM Fig 4 shows the $CH_4$ mixing ratio time series by measurement height. There was a prominent $CH_4$ peak of ~26 ppm measured at 3.2 m, which is almost double the second highest $CH_4$ measurement. This peak was measured during the second transect of the height. The peak $CH_4$ mixing ratio measured in the first transect was only

~3 ppm. The range of measured concentrations in this experiment signal that atmospheric dispersion was complex. There are several plausible dynamics. First, variable wind speeds at the source can result in variable concentrations downwind as the $CH_4$ is initially diluted into the atmosphere at the release stack. For example, a lull at the stack could have created a pocket of high $CH_4$ concentration—which was subsequently advected downwind. Second, mixing at these close ranges can be unpredictable—certain turbulence structures could maintain very high concentrations without lateral mixing. Third, the plume

moves up and down with turbulent structures. Because measurement of each height was offset in time (c.f., Rella et al., 2015), each transect could intersect different parts of the plume. It is possible that subsequent transects could 'double count' the peak of the plume, or in other cases completely miss the center of the plume. Meaningful proxies for conditions where these types of dynamics exist could include wind speed or direction variability within the experiment. Future work could tightly sync co-located wind data to precisely identify the wind signatures associated with turbulent features, modeling their impact on the

walking measurements. Further, measuring at further downwind positions (> 10 m) theoretically should reduce the strength of effects associated with variable near-source mixing. This effect is not strong, if it exists at all (Figure 8d), but may require further work to explore in more depth.

    In general, there was a lack of correlation between percent relative error and wind speed, release rate, and downwind distance, indicating that the performance of the flux plane method was not systematically affected by variations in these

variables during the experiments. Although, larger errors were associated with low release rates ≤ 0.3 kg $CH_4$/h, and negative errors were possibly more frequent at wind speeds > 6 m/s and downwind measurement distances of 10m. This phenomena of relatively constant error across the full range of release rates highlights a potential performance limitation of the method for quantifying sources emitting $CH_4$ at very low rates, as relative error would continue increasing as rates decrease.

    The test results of the human-portable flux plane method are comparable to those reported by Corbett and Smith

(2022) for their drone system, albeit with some differences between testing designs and test envelopes. Corbett and Smith (2022) performed 12 non-zero releases at generally much larger rates than our work (0 kg $CH_4$/h to 25 kg $CH_4$/h). $CH_4$ was measured mainly at 50 m or 300 m downwind of the release point. $CH_4$ sensor types (open path vs. closed path) and frequencies (10 Hz vs 1 Hz) were also different between Corbett and Smith (2022) and our work. We used the results for the non-zero



rates reported in Corbett and Smith (2022) to calculate a mean relative error of 5.6%; 68% of the estimated CH$_4$ rates in their

study were within ±36.7% of the release rates. In their study, CH$_4$ was released from a 56 m tall crane, which is different from our near-surface releases. This is noteworthy in that the experimental design in Corbett and Smith (2022) probably did not require extrapolating CH$_4$ mixing ratios measured at the lowest level with the drone to the ground and it is likely all CH$_4$ would have been captured in their flight path. Extrapolation down to the surface for quantification is required in many applications of the flux plane method, such as measuring CH$_4$ emissions from oil and gas sites with shorter equipment. Drones have a

minimum flight altitude. The results of Corbett and Smith (2022) are representative of the method's application to tall stacks or lofted plumes, but not individual oil and gas well pads with shorter equipment where plumes reside closer to the surface. Additionally, Corbett and Smith (2022) did not test any non-zero release rates <1 kg/h, where enhancements will be lower and potentially become less accurately isolated from sensor noise. We performed a total of twelve experiments at CH$_4$ release rates between 0.2 kg/h to 0.92 kg/h, and our results indicated that larger errors were associated with low release rates. Despite the

aforementioned differences, errors reported here are similar in magnitude to those reported by Corbett and Smith (2022).

The test results can also be compared with Rella et al. (2015)'s controlled release testing of the flux plane method using a vehicle-based system; however, there were differences in the method. The CH$_4$ data were collected in a vertically synchronous flux plane pattern using a mast affixed to a vehicle's front bumper and a gas sample storage manifold. Rella et al. (2015) fit a Gaussian model vertically to horizontally integrated CH$_4$ mixing ratios to extrapolate the relatively low

measurement heights upwards. This noted, comparing our results to Rella et al. (2015)'s provides additional context to the human-portable flux plane variant developed and tested in our work. Rella et al. (2015) performed 120 controlled releases at rates between 0.43 kg CH$_4$/h to 2.14 kg CH$_4$/h. The vehicle transected the controlled plumes at 5 m to 81 m downwind distances (mean 34 m). The mean release height in the study was 2.2 m, similar to our release height of ~2.4 m. Mean wind speeds (3.6 m/s) were lower in Rella et al. (2015)'s controlled testing than ours (4.8 m/s). The reported accuracy in the study was 24%,

and based on the geometric standard deviation, 67% of estimates were within a factor of 1.9 of the release rates. This translates to an uncertainty range of -47% to 90%, although it is not directly comparable to our uncertainties, as we used arithmetic rather than geometric measures of central tendency in assessing error.

The vehicle-based flux plane method—as demonstrated in Rella et al. (2015)—was also evaluated in a simulation study (Kumar et al., 2021), but overall, there are practical limitations to the heights of poles that can be affixed to vehicles and

subsequently, the heights of sources that can be measured at further downwind distances with this method.

The oil and gas industry is an ideal setting to apply the human-portable flux plane method because of the need to understand source rates that are elevated and difficult to measure with other approaches (Vollrath et al., 2024). One potential application of this method is site-scale emissions quantification, which has a wide variety of existing methods. Caulton et al. (2018) summarize uncertainty ranges from several mobile (Lan et al., 2015; Rella et al., 2015; Yacovitch et al., 2015) and

stationary (Brantley et al., 2014; Foster-Wittig et al., 2015) vehicle-based studies. Mobile vehicle-based methods, which typically use atmospheric dispersion models to estimate CH$_4$ rates based on brief downwind plume passes, generally yield much higher uncertainties (50% to 350%) than stationary vehicle-based methods (25% to 60%), which involve parking the





vehicle downwind, collecting data for ~20 minutes, and fitting a dispersion model to the data. $CH_4$ is usually measured at a single height with these methods (c.f., Rella et al., 2015). The larger uncertainties for mobile vehicle-based methods are largely
due to their reliance on brief plume snapshots, which may not represent time-averaged plume characteristics (Caulton et al., 2018). Our results suggest that the human portable pole method achieves performance somewhere in between stationary and mobile vehicle-based methods. This noted, the method cannot sample as quickly as mobile vehicle-based methods—sampling times are similar to stationary vehicle-based methods. The human portable pole method is best suited for close range, moderately high sources.

**4.3 Forward Gaussian optimization method**

We developed an alternative approach to quantify $CH_4$ emissions using data collected in the flux plane pattern in situations where measurements cannot fully capture the entirety of the plume. The experiments and calculations simulate using a telescoping pole that is slightly taller than the source and has a much higher chance of missing the upper portion of the plume.

        This alternative approach achieved similar performance in estimating release rates compared to the flux plane method
despite our removal of the top two measurement levels to test the method. The mean relative error across all estimates was 2.4% and 68% of estimates were within ±51.1% of the release rates, suggesting a slightly tighter error range and less bias compared to Rella et al. (2015)'s reported errors using a similar analytical method. Using optimization with constraints may have helped the Gaussian model successfully fit plume peaks in the vertical dimension despite temporal asynchrony to the measurement heights in this study and plume meander between the heights.

595        In some cases, $CH_4$ plumes as measured during the experiments did not closely conform to the Gaussian shape in the vertical dimension. Observed vertical plume shapes were often skewed with $CH_4$ mixing ratios falling rapidly at the second-highest or highest measurement heights. This depended on the downwind distance and wind speed during the experiments and the overall narrowness of plumes. Despite this, estimated rates with the forward Gaussian method were similar to the $CH_4$ release rates, demonstrating its efficacy as a method to use or fall back on in cases where the majority of a plume cannot be
captured with the telescoping pole.

        Removing the data acquired at the top two measurement levels during the experiments to test the forward Gaussian method simulated using a telescoping pole (~3.2 m) slightly taller than the source (~2.4 m) to measure $CH_4$. This simulated three scenarios: (i) the top measurement level aligned with the plume centerline, assuming some rise of the plume after exiting the stack; (ii) the top measurement level was slightly above the centerline, indicating the plume was sheared off the stack and
allowing partial capture of its upper half; and (iii) in some cases, the plume was mixed downward, enabling full capture of the plume closer to the source despite the limited vertical extent of measurements. The tests and results encode each of these possible scenarios. $CH_4$ plumes are highly dynamic and thus it is nearly impossible to predict which of these three scenarios will happen when performing measurements. As a best practice when using this method, a measurement kit with vertical capabilities slightly taller (e.g., 1 m) than the source or the plume's predicted average location should be used. Infrared imaging
of source plumes prior to measurements can support customizing a pole-based kit. Forward model results could improve if



more vertical information on the plume's upper half is provided to the model, provided those measurements are made within the key mass of the plume and transects meet the criterion for further data processing used in this study. Overall, the method allows the height restrictions of the flux plane method to be slightly relaxed, increasing the height of sources that can be measured, such as slightly taller (~ 7 m) tanks or other taller oil and gas equipment.

We developed and tested the forward Gaussian optimization method based on a single-source release. Thus, the method, as documented in this study, should only be applied presently to single point-sources or smaller scale "point-sources" (e.g., oil and gas well pad) where potentially multiple co-located plumes are well mixed. Deviation from this risks applying the method in situations that poorly match the method's error characterization, and actual errors could be much greater. Oil and gas facilities with large spatial footprints and small distances between equipment groups should be avoided, as

measurements at close distances would likely show several individual, poorly mixed plumes, which traditional Gaussian models cannot adequately reproduce (e.g., Nathan et al., 2015). Although, the single-source version of the Gaussian method could be used if individual plumes from a pole-based survey of a large oil and gas facility were spatially distinct enough to be isolated. A multi-source inverse version of the Gaussian plume model has been used to successfully quantify $CH_4$ emissions from oil and gas production sites (Caulton et al., 2018). Future work may investigate the efficacy and performance of a multi-

source configuration of the forward Gaussian optimization of pole-based measurements in estimating $CH_4$ emissions rates from multiple sources at closer downwind distances.

**4.4 Multiple measurements**

Despite the limited number of replicate experiments performed, our results based on averaging errors suggests that performing multiple measurements improves the accuracy of $CH_4$ rate estimates. More replicate experiments under a wider enveloping of

atmospheric conditions could confirm this (e.g., Barchyn et al., 2025). Overall, results from our replicate experiments indicate that measuring $CH_4$ emissions from sources multiple times under different conditions is valuable to improve the characterization of emissions and reduce uncertainties.

**5 Conclusion**

This study developed a new $CH_4$ emissions quantification method, adapting mass balance and forward Gaussian model

methods to work with data measured with a human-portable pole. We tested the system in in 44 individual controlled release experiments. We documented the methodology, addressing the overall poor reproducibility of the method presently based on a lack of details in the research literature. The human-portable flux plane method is a simple, accessible, and user-friendly variant of vehicle- and drone-based variants of the method for researchers. Moreover, the human portable variant has high deployment readiness in that researchers can quickly mobilize when atmospheric conditions are optimal—this operational

flexibility can yield better results.



The human-portable flux plane achieved a mean relative error of -10.1% in a series of 44 controlled release experiments with rates between 0.2 to 5.6 kg $CH_4$/h. A total of 68% of the quantified rates were within ±38.3% of the release rates, similar to the performance drone-based applications of the method (Corbett and Smith, 2022). There was a general lack of correlation between relative error and wind speed, release rate, and downwind distance. Based on our results, we estimate

that the method can be used quantify $CH_4$ emissions from sources that are about one-half the height of the telescoping pole used under moderate wind and at closer downwind distances ($\leq 30$ m). Taller sources or slightly buoyant plumes from shorter sources can be targeted in periods of higher wind speeds owing to the method's high deployment readiness and operational flexibility.

There are physical limitations to the height of a telescoping pole that a person reasonably operate. Therefore, the

traditional flux plane method—where the majority of the $CH_4$ plume must advect through the plane—is not always appropriate depending on the source, downwind distance of the measurements, or atmospheric conditions. Thus, we developed and tested an extension of the flux plane method based on forward Gaussian model optimization. The method extension uses data collected in the same pattern but is processed differently and designed to be used in cases where the traditional flux plane method cannot fully capture the plume vertically. We tested the method by removing the top two measurement heights from

the forward Gaussian optimizations. The method was slightly less accurate than the flux plane method but had less bias. The mean relative error was 2.4% and 68% of estimates were within ±51.1% of the release rates.

From this study, we recommend for the forward Gaussian method that (i) the telescoping pole must be sufficiently tall that it can measure the plume's centerline at minimum, and (ii) the spatial extent of sources such as oil and gas well pads is small enough that potentially separate plumes from different sources mix into a single "site level" plume before being

transected downwind. At present, the forward Gaussian method documented in this study cannot quantify $CH_4$ emissions from large spatial extent, multi-plume sources. Future work may improve the Gaussian optimization algorithms developed as part of this study to handle multiple sources.

The human-portable flux plane method and its extension using forward Gaussian modeling is a new addition to the $CH_4$ emissions quantification toolkit designed to target moderate-height to slightly taller sources under moderate wind speeds

and at closer downwind distances. Many of these sources are present in the oil and gas industry, but these methods can likely be applied to sources in other industries and sectors. The methods are accessible, reproducible, and rapidly deployable, which creates opportunity for researchers to independently measure, monitor, and confirm $CH_4$ emissions.

*Data availability.* The true release rates and the quantified $CH_4$ rates can be found at: [to add after peer-review]

*Supplement.* The supplementary material (SM) related to this article can be found at: [to add after peer-review]

*Author contribution.* CH conceptualized the idea of using a pole to measure $CH_4$ mixing ratio at height. CV conceptualized using the flux plane method with a pole to quantify emissions. CV, CH, TB, AM, and CW designed and performed the controlled release experiments. CV developed the flux plane and forward Gaussian plume optimization algorithms and



quantified the CH$_4$ emissions. CV analyzed the data and prepared a draft manuscript. All co-authors assisted in preparation of the final manuscript.

*Competing interests.* The authors declare they have no competing interests.

*Acknowledgements.* The authors would like to thank Joseph Samuel for his field assistance and Zhenyu Xing for providing a photograph of the controlled release setup.

*Financial support.* This project was undertaken with the financial support of the Government of Canada under the Calgary Urban Emissions Measurement Testbed project. Ce project a été réalisé avec l'appui financier du gouvernement du Canada.
Additional financial support was provided by the Alberta Methane Emissions Program (www.amep.ca) and a doctoral scholarship from the Natural Sciences and Engineering Research Council (NSERC) of Canada.

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
