# Peer review of "A Human-Portable Mass Flux Method for Methane Emissions Quantification: Controlled Release Testing Performance Evaluation"

_EGUsphere, 2025_

## Author Response (AR1)

**Response to egusphere-2025-3895 reviewers of "A Human-Portable Mass Flux Method for Methane Emissions Quantification: Controlled Release Testing Performance Evaluation"**

We thank reviewers and editors for helpful and insightful reviews that have led to improvements to the paper.

We provide responses to each comment in blue.

Line numbers in comments from editors and reviewers correspond to the original revision submission as they are copied here directly from the reviews provided by the editor.
Line number in responses refer to the updated manuscript unless otherwise noted.

**Referee comment #1:**

The manuscript presents a method to estimate methane emissions coming from moderate heights and reasonable wind speeds. As a substitute for drone or airplanes that cannot always go close to the ground, they propose a simpler and cheaper method: they use a portable instrument which is sampling from a pole.

The manuscript is well detailed with a lot of thought into the potential issues. A combination with a Gaussian plume model is also described that can help when the source is slightly higher than the pole.

I recommend publication subject to technical corrections as listed below:

We thank the reviewer for their time and effort in reviewing our paper and for the technical corrections they provided. We have made the necessary corrections which have led to improvements to the paper. Please see responses to the line-by-line comments below.

Please note that line numbers in comments from editors and reviewers correspond to the original revision submission as they are copied here directly from the reviews provided by the editor. Line number in responses refer to the updated manuscript unless otherwise noted.

p2 l51 remove "is" in "Further complicating this is some release points of interest are elevated"

Thank you. We have revised to "Further complicating this, some release points of interest are elevated (e.g., flares, tank tops, etc.), meaning the plume disperses from the release point both up and down within the atmosphere." Please see L54-55 of the revised version.

p8 l197/198 this sentence is not clear. Please clarify.

We agree with the reviewer that this sentence is unclear owing to potential confusion between "heights" and "measurement levels." We have clarified to "On 6 August 2024, the top height was 5.6 m with approximately 1 m spacing between measurement levels." See L206. We also revised L195 of the previous version to reemphasize that methane mixing ratio was measured at six measurement levels. Please see L204 of the revised version.

p17 l406-407 Please clarify the unit of time (minute and second).

Thank you. Yes, these units are min and s. We have revised. See L415.

P27 l635 remove "in"

Thanks. Correction made.

p28 l645 add 'to' in "the method can be used quantify CH4"

Thank you. Correction made.

p28 l649 add "can" in "that a person reasonably operate"

Thanks. Correction made.

**Referee comment #2:**

General comments:

Vollrath et al. present an improvement of methane walking surveys through the inclusion of a pole that allows vertical sampling. The method is tailored to quantify low emission rates of sites with small footprint which have been difficult to reliably measure with techniques like drones, aircraft or satellite. The test results should be considered a first step to validating this method, as the controlled release was conducted under ideal conditions, with no unknown other sources, a known release location with constant rate and known emission height in flat terrain and very small range of emissions. Nevertheless, this is a very informative study and addresses the issue of lower emitting sites that have been ignored in much of the literature so far. The study parameters and results are presented very clearly and the manuscript is easy to follow. The manuscript details practical constraints as well as theoretical considerations. Hence, this approach has the potential for wide adoption due to its easy of implementation.

After very few minor comments are addressed, I can wholeheartedly recommend this study for publication and expect it to be of interest to experts in the field as well as the wider readership of AMT.

We thank the reviewer for their time and effort in reviewing our paper and are grateful for their expertise which has led to improvements to the paper. We have addressed all of the reviewer's comments line-by-line below. Thank you.

Please note that line numbers in comments from editors and reviewers correspond to the original revision submission as they are copied here directly from the reviews provided by the editor. Line number in responses refer to the updated manuscript unless otherwise noted.

Line 11: The concentration range tested is very low. There are a lot of sites with higher emissions and this manuscript could add a bit more information about the fact that this is a complementary

method to vehicle and aircraft methods.
https://online.ucpress.edu/elementa/article/doi/10.1525/elementa.284/112796/Methane-emissions-from-oil-and-gas-production

We thank the reviewer for their comment and agree that we did not test rates representative of super-emitting sites (e.g., 25 kg/h to > 100 kg/h). We mention on L157 to 160 of the original version that we selected the 0.2 kg/h to 5.6 kg/h range such that the testing covered a variety of oil and gas infrastructure such as natural gas distribution facilities and lower-emitting production sites, and that the rates were not representative of higher emitting infrastructure. The pole method could be used to quantify sites emitting at higher rates, as demonstrated by the method's performance at the upper range of our tested range, but there are important considerations for exceeding the trace gas analyzer's (LiCOR 7810) measurable range of 0 ppm to 100 ppm and the effect of high-rate, high concentration plumes on the pole operator in close-range measurements. The pole method is also slow compared to mobile vehicle-based systems, aircraft, and satellites, so it cannot acquire large samples like these methods can. The pole method is useful for targeted, independent, and repeat measurements of low-emitting and moderate height sources at a close-range.

We have added more information on the complementary nature of the pole method in several locations throughout the MS, most notably in the discussion. Please see these locations in the revised version:

Abstract L16 to 19.

Discussion L596 to L614.

Conclusion L699 to 701.

Line 15: The method can not be less accurate AND have less bias, as accuracy includes biases. The method is slightly less precise and also has less bias.

Thank you. Correction made. Please see L15 and L689 of the revised version.

Line 44: How do you account for the potential change in plume share during the transect? Given that vehicles, drones and aircrafts are much faster they capture a 'snapshot' of the plume enhancements. Is this the same for walking surveys? And does this apply for larger plumes as well?

Yes, this is the same for walking surveys. Each transect is a snapshot of the plume at a given measurement height. The transects were walked at a moderate pace similar to the speed at which a quadrotor drone would transect a plume. We agree with the reviewer that vehicles and aircraft are faster.

All methods that transect a plume have issues with temporal dynamics and plume changes during the transect. This is somewhat unavoidable because it takes time to go from one side of a plume

to another. In a sense, none of the methods that use a vehicle or foot to transect the plume are truly 'snapshots' of a plume. This can be mitigated partly with our work practice of moving across the plume and never, ever stopping in the plume. Running could be an improvement, but this is not realistic because control of the pole suffers when running, and there are health and safety concerns associated with running with a long pole.

Broadly, we acknowledge that temporal changes in the plume likely occur during transects, but we suspect the impact is minimal and will form a part of error statistics.

Atmospheric stability and the overall wind speed, which affect how a plume disperses and thus influence its shape, typically change over longer time scales than the scale of the experiments in our study (10 min to 20 min). At the temporal and spatial scales (10 m to 30 m downwind) of the pole measurements, variability in the plume's characteristics are primarily governed by gusts and lulls in the wind. Variability in the wind's speed and direction on the order of tens of seconds can change the plume's location horizontally and vertically between transects.

Our analysis and work practice has some mitigating factors. First, we walked longer transects around the release stack and used the odor of mercaptan to ensure that the plume was fully transected horizontally (L173 of the original version). In the vertical dimension, we used a pole that was about 2 to 2.3 times the height of the release stack, which was tall enough to capture the plume in close range surveys (10 m to 30 m) while accounting for vertical plume meander. Wind speeds were also 3 m/s or greater in all experiments enabling full capture of the plume vertically at these downwind distances. The pole measurements should not be performed when there is no wind and the atmosphere is highly unstable. Second, with the flux plane method, the measurements are projected and interpolated onto a 2D grid. The interpolation smooths out this variability in the plume's location across the dimensions of the measurements. For the Gaussian method, we addressed changes in the plume's horizontal location between transects by identifying the plume's centerline in each transect (peak mixing ratio) and then aligning the centerlines of all transects, similar to Caulton et al. (2018)'s method for multi-pass Gaussian plume measurements with a vehicle-based system. Once aligned, the transects were ingested by the customized Gaussian plume model, where an optimizer solved for the optimal horizontal and vertical dispersion parameters, stack height, and emissions rate required to predict the closest concentrations to those measured. The transects could not be aligned in the vertical dimension for the Gaussian plume model because this would reduce the measurements to a single height. Thus, experiments with less plume meander in the vertical dimension tended to produce the best model fits.

Plume dispersion increases vertically and horizontally away from the source as the downwind distance increases. The same concepts discussed in this response apply to larger plumes (i.e., those measured further downwind). At farther downwind distances (e.g., 50 m) than those at which measurements were performed in this work (10 m to 30 m), the pole may miss some of the upper portion of the plume unless a taller pole is used, wind speeds are higher, or both. Modeling

approaches—such as the customized Gaussian method demonstrated in this work—can help in situations when the pole cannot fully capture the plume vertically. There is a threshold where plumes may become so large in both dimensions—such as with landfills or other very large area sources—that the human-portable pole method may become impractical or unsuitable and vehicle-based or other methods are more suitable to measure emissions. We have added discussion on this on L513 to 518.

References

Caulton, D. R., Li, Q., Bou-Zeid, E., Fitts, J. P., Golston, L. M., Pan, D., Lu, J., Lane, H. M., Buchholz, B., Guo, X., McSpiritt, J., Wendt, L., and Zondlo, M. A.: Quantifying uncertainties from mobile-laboratory-derived emissions of well pads using inverse Gaussian methods, Atmos. Chem. Phys., 18, 15145–15168, https://doi.org/10.5194/acp-18-15145-2018, 2018.

Line 47: Please consider using "tens" and "hundreds" instead of 10s and 100s to improve readability and avoid confusion with 10s, i.e. 10 seconds.

Thank you. Correction made. We have revised throughput the manuscript as well.

Line 95: Indeed, drones can not be flown everywhere, but walking surveys can also not be performed everywhere, this should also be recognized here.

We thank the reviewer for their comment and agree. We believe mention of this fits better in the following paragraph, as we have not yet introduced walking measurements on this line. We have added the sentence "Similar to how drones cannot be flow everywhere, it should be emphasized that the walking measurements demonstrated in this study also cannot be performed everywhere—access, terrain, source characteristics, safety, and meteorological conditions can create constraints for the method." Please see L105 to 108.

Line 97: Significant controlled release studies have been conducted in Canada funded by EREF, which seem pertinent to cite here to judge the relative error reported here. (Report can be found here: EREF-Funded Study Highlights Advances in Measuring Landfill Methane Emissions | EREF)

Thanks for the comment. We have revised and mentioned this study on L96 to L99 in the introduction. Additionally, we have added in the discussion on L513 to 518 that the pole method may not be practical or suitable to quantify emissions from very large area sources like landfills. Given the scale of these sources, the wind would likely advect the majority of the methane molecules emitted above the top of the pole, and long walking distances would be required. The methods demonstrated in our study can be used with drones to quantify emissions from landfills or other large area sources, as demonstrated by the EREF study mentioned by the reviewer. We mention that study in these lines too.

Line 185 – Figure 2: The figure suggests that the farther transect is far from semi-circles around the site, but are definitely long enough to capture the plumes.

Thank you. We have changed semi-circle to "arc" throughout the manuscript.

Line 415/465: The linear regression fit should not be forced through the origin. In the real-world unknown sources could add an offset on the measured results. Forcing fits to the origin can make the correlations artificially high. However, the data here is obviously very highly correlated and the forced fit might not cause large differences.

The regression fit is forced through the origin because in controlled release experiments when gas is not being released methods should quantify a rate of 0 kg/h. We agree with the reviewer that this is not always the case, as it can be challenging to disambiguate between emissions from target and non-target sources in some cases, or target sources and background especially if the background is highly variable. The "offset" the reviewer is referring to arises when detection and quantification algorithms cannot reliably separate the methane associated with the target plume from that of other sources or the background—some of the non-target methane ends up being included in rate estimates.

The Carbon Management Canada Field Research Station where we performed the controlled release tests is a generally clean environment for non-target $CH_4$ sources by design. We own the compressed natural gas (CNG) trailer used for the releases to further control the environment by enabling relocation of the releases to address any unexpected non-target plumes. Furthermore, we screened the release setup and the surrounding area with multiple methods such as an OGI camera and a vehicle-based system (Barchyn et al., 2025), as mentioned in the manuscript, to ensure the experiments were not affected by non-target sources. Finally, in our experiments, the length of the arc transects enabled acquiring a large number of measurements outside of plumes to use in approximating background.

If we tested a zero release—which was outside of the scope of this work but is an opportunity for future research—we believe the data collection and processing procedures demonstrated in this study would have led to an estimated rate of zero. Non-target sources did not present an issue in the experiments given the clean environment, which was confirmed. If a method is quantifying a non-zero rate when the release is zero, then it suggests that the method's quantification skill could be improved, with emphasis on estimating background, or the environment was unclean. Overall, these examples justify forcing the fit through the origin for our study, which is a common approach in controlled release studies when the surrounding environment is clean (e.g., Barchyn et al., 2025).

References

Barchyn, T., Clements, M., Gough, T., Hugenholtz, C., Munn, A., Samuel, J., Wearmouth, C., and Vollrath, C.: PoMELO Passive Blind Test Results: Emissions detection and quantification, https://doi.org/10.31223/X54Q65, 21 February 2025.

Line 657: The value of replicate measurements had been discussed in previous studies using mobile systems: https://amt.copernicus.org/articles/18/3569/2025/

We thank the reviewer for their comment and request clarification regarding the line number referenced here. We believe the reviewer is referring to L627 of the original version (Section 4.4.), where we discuss the replicate measurements.

We have added mention of this study in the paragraph in this section. "This aligns with findings from other recent studies that performed multiple plume transects in controlled release tests evaluating the quantification performance of mobile systems (vehicles) for different applications (Barchyn et al., 2025; Tettenborn et al., 2025)." Please see L660 to 664 of the revised version.